# HPV Vaccination after Primary Treatment of HPV-Related Disease across Different Organ Sites: A Multidisciplinary Comprehensive Review and Meta-Analysis

**DOI:** 10.3390/vaccines10020239

**Published:** 2022-02-04

**Authors:** Violante Di Donato, Giuseppe Caruso, Giorgio Bogani, Eugenio Nelson Cavallari, Gaspare Palaia, Giorgia Perniola, Massimo Ralli, Sara Sorrenti, Umberto Romeo, Angelina Pernazza, Alessandra Pierangeli, Ilaria Clementi, Andrea Mingoli, Andrea Cassoni, Federica Tanzi, Ilaria Cuccu, Nadia Recine, Pasquale Mancino, Marco de Vincentiis, Valentino Valentini, Gabriella d’Ettorre, Carlo Della Rocca, Claudio Maria Mastroianni, Guido Antonelli, Antonella Polimeni, Ludovico Muzii, Innocenza Palaia

**Affiliations:** 1Department of Maternal and Child Health and Urological Sciences, Sapienza University of Rome, Policlinico Umberto I, 00161 Rome, Italy; violante.didonato@uniroma1.it (V.D.D.); giorgio.bogani@uniroma1.it (G.B.); giorgia.perniola@uniroma1.it (G.P.); sara.sorrenti@uniroma1.it (S.S.); federica.tanzi@uniroma1.it (F.T.); ilaria.cuccu@uniroma1.it (I.C.); nadia.recine@uniroma1.it (N.R.); pasquale.mancino@uniroma1.it (P.M.); ludovico.muzii@uniroma1.it (L.M.); innocenza.palaia@uniroma1.it (I.P.); 2Department of Public Health and Infectious Diseases, Sapienza University of Rome, Policlinico Umberto I, 00161 Rome, Italy; eugenionelson.cavallari@uniroma1.it (E.N.C.); gabriella.dettorre@uniroma1.it (G.d.); claudio.mastroianni@uniroma1.it (C.M.M.); 3Department of Oral and Maxillofacial Sciences, Sapienza University of Rome, Policlinico Umberto I, 00161 Rome, Italy; gaspare.palaia@uniroma1.it (G.P.); umberto.romeo@uniroma1.it (U.R.); andrea.cassoni@uniroma1.it (A.C.); marco.devincentiis@uniroma1.it (M.d.V.); valentino.valentini@uniroma1.it (V.V.); antonella.polimeni@uniroma1.it (A.P.); 4Department of Sense Organs, Sapienza University of Rome, Policlinico Umberto I, 00161 Rome, Italy; massimo.ralli@uniroma1.it; 5Department of Radiological, Oncological and Pathological Sciences, Sapienza University of Rome, Policlinico Umberto I, 00161 Rome, Italy; angelina.pernazza@uniroma1.it (A.P.); carlo.dellarocca@uniroma1.it (C.D.R.); 6Department of Molecular Medicine, Sapienza University of Rome, Policlinico Umberto I, 00161 Rome, Italy; alessandra.pierangeli@uniroma1.it (A.P.); guido.antonelli@uniroma1.it (G.A.); 7Department of Emergency, Sapienza University of Rome, Policlinico Umberto I, 00161 Rome, Italy; ilaria.clementi@uniroma1.it; 8Department of Surgery “Pietro Valdoni”, Sapienza University of Rome, Policlinico Umberto I, 00161 Rome, Italy; andrea.mingoli@uniroma1.it

**Keywords:** human papillomavirus, HPV, vaccination, cervical cancer, vulvar cancer, anogenital warts, laryngeal papillomatosis

## Abstract

Objective: To assess evidence on the efficacy of adjuvant human papillomavirus (HPV) vaccination in patients treated for HPV-related disease across different susceptible organ sites. Methods: A systematic review was conducted to identify studies addressing the efficacy of adjuvant HPV vaccination on reducing the risk of recurrence of HPV-related preinvasive diseases. Results were reported as mean differences or pooled odds ratios (OR) with 95% confidence intervals (95% CI). Results: Sixteen studies were identified for the final analysis. Overall, 21,472 patients with cervical dysplasia were included: 4132 (19.2%) received the peri-operative HPV vaccine, while 17,340 (80.8%) underwent surgical treatment alone. The recurrences of CIN 1+ (OR 0.45, 95% CI 0.27 to 0.73; *p* = 0.001), CIN 2+ (OR 0.33, 95% CI 0.20 to 0.52; *p* < 0.0001), and CIN 3 (OR 0.28, 95% CI 0.13 to 0.59; *p* = 0.0009) were lower in the vaccinated than in unvaccinated group. Similarly, adjuvant vaccination reduced the risk of developing anal intraepithelial neoplasia (*p* = 0.005) and recurrent respiratory papillomatosis (*p* = 0.004). No differences in anogenital warts and vulvar intraepithelial neoplasia recurrence rate were observed comparing vaccinated and unvaccinated individuals. Conclusions: Adjuvant HPV vaccination is associated with a reduced risk of CIN recurrence, although there are limited data regarding its role in other HPV-related diseases. Further research is warranted to shed more light on the role of HPV vaccination as adjuvant therapy after primary treatment.

## 1. Introduction

Prophylactic human papillomavirus (HPV) vaccines are considered to be the most successful and cost-effective public health measure to prevent HPV infection and related cancers across different organ sites [1]. In 2006, the Food and Drug Administration (FDA) approved the first HPV vaccine designed to prevent HPV-related cancer (Gardasil^®^). The vaccine was initially approved for women, and then expanded also to men in 2009. These vaccines consist of noninfectious, HPV-like particles that elicit the production of neutralizing L1-specific antibodies blocking the viral entry into host cells. Currently, there are three types of HPV vaccines available: 2-valent (Cervarix^®^), 4-valent (Gardasil^®^), and 9-valent (Gardasil9^®^), all targeting the two most oncogenic serotypes, HPV 16 and HPV 18 [2,3]. According to the updated recommendations of the Advisory Committee on Immunization Practices (ACIP), HPV vaccination is intended for both females and males aged 9 to 26 years [4]. All HPV vaccines are highly immunogenic, with more than 98% of recipients developing antibodies within one month after completing vaccination, and they seem to provide protection for at least 10 years [5].

Despite the remarkable impact on public health outcomes worldwide, HPV vaccination is not currently recommended for older adults or those with prior HPV exposure, leaving a large portion of the population at risk for HPV-related diseases [6]. In particular, the following categories could benefit from HPV vaccination beyond current recommendations: (1) adults who did not fulfill age inclusion criteria when the first HPV vaccine was introduced, thus being excluded from free vaccination programs; (2) individuals who did not receive or complete vaccination, albeit eligible, either because HPV vaccines were not available, such as in developing countries, or out of their personal choice (since HPV vaccination is not mandatory as for other vaccines); (3) rare cases of failure to achieve immunization after vaccination; (4) vaccinated adults who gradually lose long-term immunization (probably starting 10 years after vaccination); (5) individuals already exposed to prior HPV infection. HPV infection may lead to subclinical and transient, latent, or clinically relevant diseases. HPV-mediated diseases tend to recur frequently, and this risk is consistent with either new infections, auto-inoculation across different organ sites, or episodic reactivations of preexistent latent infections [7].

As no vaccine has yet been licensed for therapeutic use, particular interest has been raised for the putative role of prophylactic HPV vaccination as an adjuvant treatment for patients with recurrent HPV-related diseases. The rationale behind the efficacy of HPV vaccines for secondary prevention remains unclear. Since viral antigens are not exposed on the surface of infected cells, and become untargetable by antibodies after the cell entry, HPV vaccines should not be efficient in eradicating pre-existing infections [8,9]. Several hypotheses have been proposed so far to explain the exact protective mechanisms of HPV vaccination in infected individuals: (a) cross-protection towards other HPV types [10,11]; (b) the surgical treatment of HPV lesions may reduce the local inflammatory response and recover an HPV-naïve microenvironment where the vaccine might be effective [12,13]; (c) HPV vaccines stimulate cell-mediated immunity, which may also play a role in preventing recurrent infection [14]; the prevention of auto-inoculation across new exposed anatomic sites. In particular, with regard to this latter hypothesis, it should be mentioned that the new emerging concept of HPV is a commensal component of the human virome. Indeed, the ubiquity and wide diversity of high-risk HPV genotypes being proven in samples from the vagina, skin, or gut microbiota of healthy subjects, reinforces a possible mechanism of secondary prevention of HPV auto-inoculation across different sites [15,16].

No large, multidisciplinary, clinical trials have investigated the efficacy of the HPV vaccine for secondary prevention in patients with active HPV-related diseases. However, emerging data have been suggesting a putative post-expositional role for HPV vaccines, warranting additional investigation. The present systematic review and meta-analysis summarizes the currently available data on the efficacy of adjuvant HPV vaccination for secondary prevention in patients with active HPV-related diseases.

## 2. Materials and Methods

### 2.1. Search Strategy

The authors performed a literature review up to January 2022 for all English-language studies reporting the efficacy of HPV vaccination as an adjunct to standard treatment for the secondary prevention of HPV-related preinvasive diseases. HPV-related diseases include anogenital warts (AGWs), cervical cancer and cervical intraepithelial neoplasia (CIN), vulvar cancer and vulvar intraepithelial neoplasia (VIN), vaginal cancer and vaginal intraepithelial neoplasia (VAIN), anal cancer and anal intraepithelial neoplasia (AIN), penile cancer and penile intraepithelial neoplasia (PeIN), recurrent respiratory papillomatosis (RRP), and head and neck diseases.

PubMed, Scopus, Cochrane library, and clinicaltrials.gov were searched using a Boolean search algorithm for studies published up to January 2022. The following search terms and their MESH terms were used: “cervical intraepithelial neoplasia”, “vulvar intraepithelial neoplasia”, “vaginal intraepithelial neoplasia”, “anogenital warts”, “anal intraepithelial neoplasia”, “penile intraepithelial neoplasia”, “respiratory papillomatosis”, “head and neck disease”, “human papillomavirus”, “HPV”, “vaccine” (Appendix A).

Additional screening was performed of the reference lists from the relevant literature. Article abstracts and, where appropriate, full text of articles and cross-referenced studies identified from retrieved articles were screened for pertinent information. All duplicate records were removed. The overall search strategy was performed using PRISMA (Preferred Reporting Items for Systematic Reviews and Meta-Analyses) guidelines [17].

### 2.2. Study Selection and Methodologic Quality Assessment

The selection of the studies was performed independently by two authors (G.C., G.B.). Publications were evaluated dependent on predefined inclusion and exclusion criteria. Inclusion criteria were as follows: (1) randomized controlled, prospective or retrospective observational studies; (2) patients undergoing standard treatment for HPV-related disease; (3) prophylactic HPV vaccination (either shortly before or after surgery) versus no vaccination; (4) histologically confirmed HPV-related disease. The following exclusion criteria were adopted: (1) case reports, editorials, reviews and short communications; (2) studies using new HPV vaccines without Food and Drug Administration (FDA) approval; (3) absence of the unvaccinated control group; (4) studies enrolling individuals with invasive disease or immunological disorders during pregnancy.

Data extraction from each included study was performed on the basis of study characteristics and predefined outcome variables. The following variables were retrieved from each study: year of publication, study design and setting, endpoints, treatment (surgery (cold knife, CO_2_ laser, and electrosurgical), cryotherapy, radiofrequency microdebridement, intralesional antiviral injection), HPV vaccine (2-, 4-, or 9-valent), vaccination timing (before or after surgery), follow up, disease recurrence, time to recurrence. Discrepancies were resolved by discussion.

The methodological quality assessment was performed following the Cochrane Handbook for the Systematic Reviews of Interventions v.5.1.0 [18].

### 2.3. Primary Outcomes

The primary outcomes were the disease recurrence rates, both irrespective of HPV type and HPV16/18-related. Outcomes were selected and extrapolated from the studies:Cervical intraepithelial neoplasia (CIN) recurrence;Anogenital warts (AGWs) recurrence;Vulvar intraepithelial neoplasia (VIN)/Vaginal intraepithelial neoplasia (VaIN) recurrence;Anal intraepithelial neoplasia (AIN) recurrence;Recurrent respiratory papillomatosis (RRP) recurrence;Penile intraepithelial neoplasia (PeIN) recurrence;Head and neck HPV-related disease recurrence.

### 2.4. Statistical Analysis

A meta-analysis of aggregate data was performed to generate a pooled estimate using effect estimates of individual studies reported in the published literature. The data were analyzed using RevMan software (Review Manager version 5.4, the Cochrane Collaboration). Dichotomous outcomes from each study were expressed as an odds ratio (OR) with a 95% confidence interval (CI). Heterogeneity between studies was reported with the I^2^ statistic. A “hybrid” Mantel–Haenszel random-effects model with inverse-variance weighting was used in meta-analyses if any heterogeneity was detected, whereas a fixed-effect model was used if no heterogeneity was identified [19]. A value of *p* < 0.05 was considered statistically significant. We decided to examine publication bias with Egger’s test and funnel plots if the number of studies was 10 or above, since these analyses are underpowered otherwise. Six domains were evaluated: random sequence generation; allocation concealment; blinding of outcome assessor; completeness of outcome data reporting; selective outcome reporting; and other potential sources of bias.

## 3. Results

### 3.1. Study Characteristics

The systematic search resulted in 55 relevant studies (Figure 1).

Among them, 39 were excluded as they did not provide adjuvant HPV vaccination, or there was no unvaccinated control group. Sixteen studies fulfilled the predefined inclusion criteria. The main details of the included articles are shown in Table 1.

### 3.2. Risk of Bias

The risk of bias was assessed, and is detailed in Figure 2.

### 3.3. Effects of Interventions

#### 3.3.1. Cervical Intraepithelial Neoplasia Recurrence

Twelve studies were published between 2012 and 2021 [20,21,22,23,24,25,26,27,28,29,30,31]. Three were prospective non-randomized studies [24,27,29], two were randomized controlled trials [25,31], four were retrospective studies [21,26,28,30], and three were post-hoc pooled analyses of randomized clinical trials [20,22,23]. The women included in the studies were between the years of 15 and 89. The median follow-up time across the studies ranged from 2 to 5 years. HPV vaccination was administered after surgical treatment in ten studies, while either shortly before or after in two studies. The HPV vaccine was 4-valent (against HPV 6/11/16/18 genotypes) in five studies [20,21,24,25,31] and bivalent (against HPV 16/18 genotypes) in two [22,23], while five studies administered both vaccines [26,27,28,29,30].

All studies evaluated the recurrence of CIN 2+ within 6–60 months after treatment. Of the 21,472 women included in the pooled analysis, CIN 2+ occurred in 1098 women (5.1%). Heterogeneity for this comparison was I^2^ 65% (95% CI 35.5–81.1%). The pooled estimated odds ratio (OR) was 0.33 (95% CI 0.20 to 0.52; *p* < 0.0001) (Figure 3).

The subgroup analysis according to the study design, prospective versus retrospective, confirmed a lower rate of CIN 2+ recurrence in the vaccinated compared to the unvaccinated group. Heterogeneity for this comparison in prospective trials was I^2^ 77% (95% CI 44.2–90.5%). The pooled estimated OR in prospective trials was 0.31 (95% CI 0.14 to 0.72; *p* = 0.006) (Figure 4).

Eight studies [20,21,22,23,24,25,28,31] for a total of 3617 patients (1747 in the vaccinated and 1870 in the unvaccinated cohort), reported the CIN 1+ recurrence within 6–48 months after surgery. The CIN 1+ recurrence occurred in 381 women (10.5%): 136 (7.8%) in the vaccinated and 245 (13.1%) in the unvaccinated cohort. Heterogeneity for this comparison was I^2^ 73% (95% CI 44.9–86.8%). The pooled estimated OR was 0.45 (95% CI 0.7 to 0.73; *p* = 0.001) (Figure 5).

Moreover, a sensitivity analysis was performed according to the study design dividing prospective and retrospective studies, confirming a lower rate of CIN 1+ recurrence in the vaccinated compared to the unvaccinated group. Heterogeneity for this comparison in prospective trials was I^2^ 0% (95% CI 0–89.6%). The pooled estimated OR in prospective trials was 0.23 (95% CI 0.14 to 0.37; *p* < 0.0001) (Figure 6).

Two studies [20,31] for a total of 1143 patients (517 in the vaccinated and 626 in the unvaccinated cohort), evaluated the CIN 3 recurrence within 6–48 months after surgery. The CIN 3 recurrence occurred in 48 women (4.2%): 15 (2.9%) in the vaccinated and 33 (5.3%) in the unvaccinated cohort. Heterogeneity for this comparison was I^2^ 0% (95% CI 0–90%). The pooled estimated OR was 0.28 (95% CI 0.13 to 0.59; *p* = 0.0009) (Figure 7).

#### 3.3.2. Anogenital Warts Recurrence

Two studies reported on the recurrence of AGWs: Joura et al. (2012) [20], a monocentric retrospective study in women, and Coskuner et al. (2014) [32], a monocentric prospective randomized study in men. The median follow-up time across the studies ranged from 1 to 4 years. HPV vaccination was administered after surgical treatment. The HPV vaccine type was 4-valent (against HPV 6/11/16/18 genotypes) in both studies.

The two studies with a total of 656 patients (225 in the vaccinated and 431 in the unvaccinated group) evaluated the recurrence of AGWs within four years after surgical treatment. AGWs recurred in 123 women (18.8%): 55 (24.4%) in the vaccinated and 68 (15.8%) in the unvaccinated group. Heterogeneity for this comparison was I^2^ 0% (95% CI 0–90%). The pooled estimated OR was 1.04 (95% CI 0.65 to 1.65; *p* = 0.88) (Figure 8).

#### 3.3.3. Vaginal or Vulvar Intraepithelial Recurrence

Two studies [20,33] reported on the VIN/VaIN recurrence: Joura et al. (2012), a monocentric retrospective study, and Ghelardi et al. (2021), a monocentric prospective non-randomized study. The women included in the studies were between the years of 15 and 45. The median follow-up time across the studies ranged from 1 to 7 years. HPV vaccination was administered after surgical treatment. The HPV vaccine was 4-valent (against HPV 6/11/16/18 genotypes) in both studies.

The two studies with a total of 740 patients (251 in the vaccinated and 489 in the unvaccinated group) evaluated the VIN/VaIN recurrence within seven years after surgical treatment. AGWs recurred in 114 women (15.4%): 36 (14.3%) in the vaccinated and 78 (15.9%) in the unvaccinated cohort. Heterogeneity was I^2^ 44% (95% CI 0–89.6%) and the pooled estimated OR was 0.81 (95% CI 0.42 to 1.55; *p* = 0.52) (Figure 9).

#### 3.3.4. Anal Intraepithelial Neoplasia Recurrence

Only one monocentric, retrospective study (Swedish et al., 2012) [34] reported data on the high-grade AIN recurrence in men who have sex with men after 4-valent (against HPV 6/11/16/18 genotypes) HPV vaccination with 2 years of follow-up.

The study evaluated a total of 202 patients: 88 in the vaccinated group and 114 in the unvaccinated cohort showing a statistically significant reduction of AIN recurrence in vaccinated women (12; 13.6%) compared with unvaccinated (35: 30.7%). This study suggested that adjuvant HPV vaccination after surgical treatment for AIN significantly reduced the risk of disease recurrence (*p* = 0.005).

#### 3.3.5. Recurrent Respiratory Papillomatosis

One monocentric, retrospective study (Mauz et al., 2018) [35] reported on the RRP and included a total of 24 patients: 11 male and 13 female. Among the male patients, 1 had juvenile RRP and 12 had adult RRP, while, among the women, this figure was 3 and 8, respectively. The median follow-up was 22 years. HPV vaccination was administered after standard treatment (radiofrequency microdebridement and intralesional antiviral injection). The HPV vaccine was 4-valent (against HPV 6/11/16/18 genotypes).

Of the 24 patients included, 13 were in the vaccinated group and 11 in the unvaccinated group. Respiratory papillomatosis recurred in 13 patients (out of 24; 54.2%): 2 (out of 13; 15.4%) in the vaccinated and 11 (100%) in the unvaccinated cohort. The study suggested that adjuvant HPV vaccination after standard treatment for RRP significantly reduced the risk of disease recurrence (*p* = 0.004).

#### 3.3.6. Other Outcomes

No data regarding the remaining outcomes were reported in the literature.

## 4. Discussion

The HPV vaccines given before initiating sexual activity have been largely demonstrated to reduce the risk of getting infected and developing HPV-related disease. However, it is still controversial whether they are useful for infected patients with prior HPV exposure. New emerging data has highlighted that HPV vaccination might have a beneficial role in the adjuvant setting.

The best available evidence regards the prevention of recurrent CIN after surgical treatment [36,37,38]. A recent meta-analysis [39], including 11 studies and 21,310 patients, demonstrated that providing HPV vaccine as an adjunct to conization for CIN reduces the risk of recurrence. The present meta-analysis added another study (Karimi et al., 2020) [31] with 312 more patients to the previous analysis by Di Donato et al. [39], and confirmed that HPV vaccination reduces the risk of CIN recurrence. On the other hand, our systematic research revealed that there are scant data regarding the other HPV-related diseases. In particular, two studies [20,33] reported on the VIN/VaIN recurrence with no significant results. Two studies [20,32] reported on the recurrence of AGWs, again with no significant results. One study [34] reported on the AIN recurrence, and one study [35] on the RPP, both demonstrating that HPV vaccination after standard treatment significantly reduced the risk of disease recurrence. Finally, no data were found regarding PeIN and head and neck diseases. Further investigation, therefore, is warranted for these noncervical organ sites. In particular, head and neck HPV-related diseases represent a peculiar entity since, unlike other organ sites, they present only as invasive diseases, and there are no pre-invasive lesions that can be monitored [40]. They are a separate tumor entity, representing around 25% of head and neck cancers (HNC), and carry a better prognosis than squamous cell cancers, which are associated with the classic risk factors of alcohol and tobacco [41,42]. Prophylactic HPV vaccination lowers the incidence of premalignant lesions of the anogenital tract, and might also reduce the incidence of HPV-associated HNC; thus, extending the recommendation for vaccination to men has become highly recommended [43,44]. Treatment with therapeutic HPV vaccines is a promising and seemingly safe strategy for patients with HPV-positive HNC, but further prospective research is required to draw any further conclusions regarding tumor response and survival outcomes [45].

Moreover, in this intriguing scenario, some relevant points need to be addressed. In particular, the optimal timing for vaccination remains to be clarified, although it has not demonstrated to have any significant influence on the recurrence rate so far. Future research is needed to address the most appropriate timing for HPV vaccine administration, which could probably be within 30 days from the standard treatment.

Compared to the recent systematic review (2017) by Dion et al. [46] addressing the role of adjuvant HPV vaccination for the secondary prevention of active clinical HPV-related disease across different disciplines, our meta-analysis included only comparative studies providing the unvaccinated control cohort. Dion et al. found 12 relevant studies for a total of 2616 patients, and 9 of these studies demonstrated decreased disease recurrence, decreased disease burden, or increased intersurgical intervals.

Confirming the efficacy of the HPV vaccine also in the secondary prevention setting would pave the way to a new era in the management of the spectrum of HPV-related diseases. HPV-related diseases represent a substantial health burden worldwide. Persistent HPV infection causes up to 4.5% (640,000 cases) of all new cancer cases worldwide [47,48]. Moreover, they tend to frequently recur either because of new HPV infections, transient reactivations of latent infections, or auto-inoculation in different susceptible organ sites. Currently, there is a lack of screening and treatment guidelines for patients with noncervical HPV-related diseases, and health professionals need to be trained to appropriately evaluate these patients. Recurrences have been associated with decreased quality of life and significant morbidity, due to disfiguring tissue removal and loss of function [49,50]. In addition, the follow-up and treatment of HPV-related preinvasive diseases can be anxiety-provoking and expensive. Therefore, although prophylactic HPV vaccination has dramatically reduced the incidence of HPV-related diseases, there is still an unmet need to reduce the risk of recurrence of preexisting conditions in older populations.

Administering the HPV vaccine shortly before or after the standard treatment is a simple and safe intervention with potentially extraordinary outcomes. Last but not least, the therapies for recurrent HPV-related diseases are costly. In 2012, Chesson et al. estimated that HPV-related disease treatments cumulatively account for nearly $8 billion in direct health care costs per year [51]. Therefore, investing in the HPV vaccination for secondary prevention could represent a cost-effective approach in both the short and longer term, that can contribute to improvements in health outcomes at lower and more sustainable costs, while supporting universal health coverage.

Awaiting more consolidated data on these specific points, it should be acknowledged that our meta-analysis has several strengths and limitations. The strengths include the following: (a) a comprehensive evaluation of all currently available data on HPV-related diseases across different specialties providing a large sample size; (b) the quality of the methodology assessment and the strict inclusion criteria. The limitations include the following: (a) heterogeneity between studies in terms of inclusion criteria and methodologies; (b) the analysis of both randomized and non-randomized studies; (c) selection and information bias.

## 5. Conclusions

The present systematic review and meta-analysis demonstrates that adjuvant HPV vaccination is associated with a reduced risk of CIN recurrence, while reporting scant data regarding its role in other HPV-related diseases. Further randomized trials are needed to shed more light on the post-expositional role of HPV vaccines across different disciplines and potentially drive post-expositional HPV vaccination into daily practice. Rediscovering the role of prophylactic HPV vaccines in the secondary prevention setting could pave the way to a new era in the management of HPV-related diseases.

## Figures and Tables

**Figure 1 vaccines-10-00239-f001:**
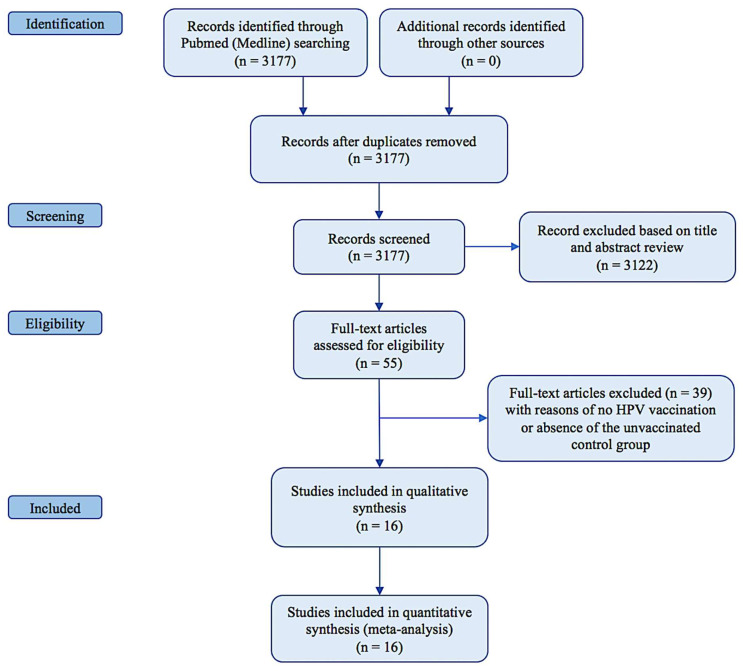
PRISMA diagram.

**Figure 2 vaccines-10-00239-f002:**
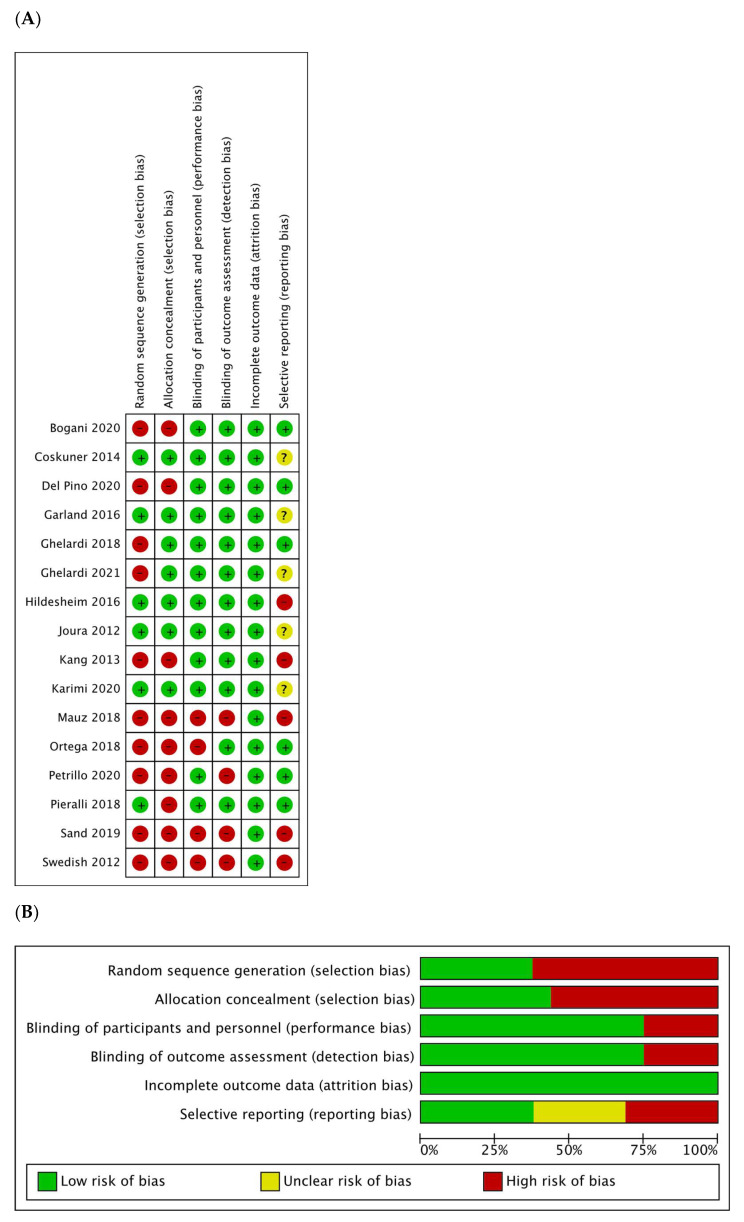
(**A**) Risk of bias summary: authors’ judgments about each risk of bias item for each included study. (**B**) Risk of bias graph: authors’ judgments about each risk of bias item presented as percentages for all included studies.

**Figure 3 vaccines-10-00239-f003:**
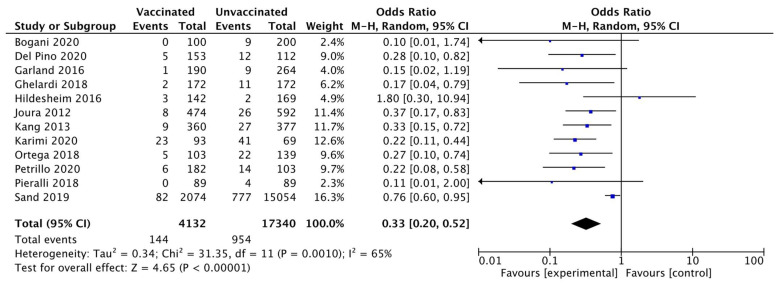
Forest plot of comparison: CIN 2+ recurrence.

**Figure 4 vaccines-10-00239-f004:**
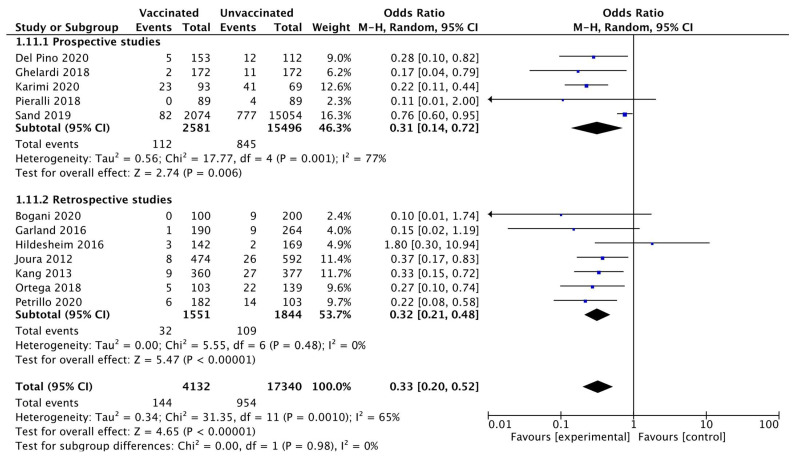
Forest plot of comparison: subgroup analysis related to the study design for CIN 2+ recurrence.

**Figure 5 vaccines-10-00239-f005:**
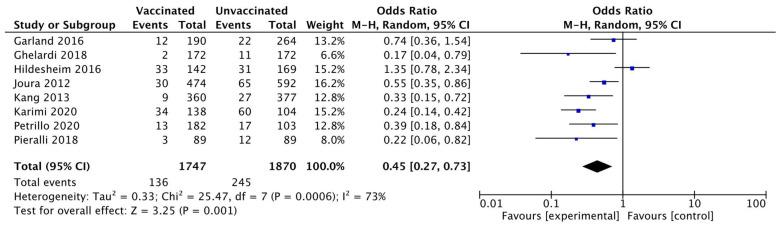
Forest plot of comparison: CIN 1+ recurrence.

**Figure 6 vaccines-10-00239-f006:**
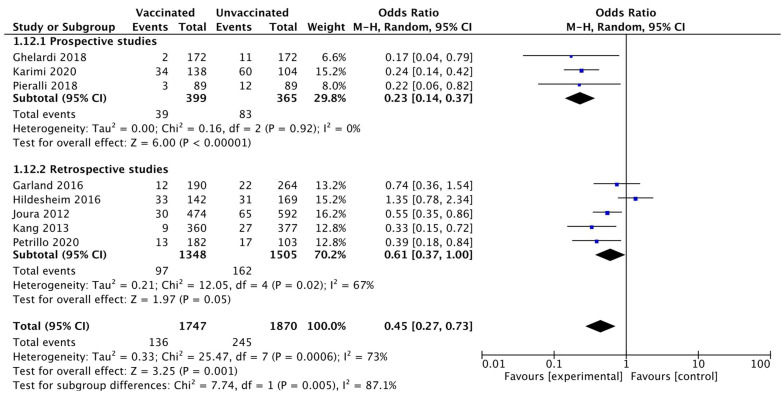
Forest plot of comparison: subgroup analysis related to the study design for CIN 1+ recurrence.

**Figure 7 vaccines-10-00239-f007:**
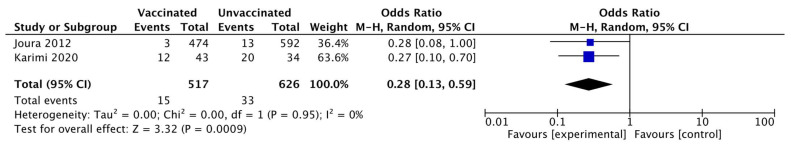
Forest plot of comparison: CIN 3+ recurrence.

**Figure 8 vaccines-10-00239-f008:**
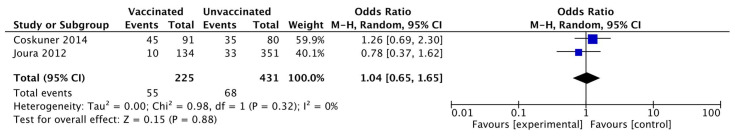
Forest plot of comparison: AGWs recurrence.

**Figure 9 vaccines-10-00239-f009:**
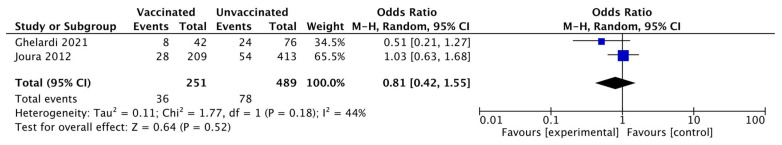
Forest plot of comparison: VIN/VaIN recurrence.

**Table 1 vaccines-10-00239-t001:** Description of the studies included.

Study, Year	Study Design	N. of Patients Age (Years)	Primary Endpoint (Recurrence)	HPV Vaccine Type and Time of Vaccination	Standard Treatment
CIN					
Joura et al., 2012 [20]	Post-hoc-pooled analysis of 2 RCT (FUTURE I and II) Follow-up 2.5 years (median)	1066 15–26	CIN 1+ CIN 2+ CIN 3	4-valent at day 1, month 2, and month 6 after surgery	LEEP (84.7%), cervical conization (12.5%), cryotherapy (0.7%), and other NA (2.1%)
Kang et al., 2013 [21]	Retrospective case-control Follow-up 3.5 years (median)	737 20–45	CIN 1+ CIN 2+	4-valent at week 1, month 2, and month 6 after surgery	LEEP
Garland et al., 2016 [22]	Post-hoc analysis of an RCT (PATRICIA) Follow-up 4 years	454 15–25	CIN 1+ CIN 2+	2-valent at months 0, 1, and 6 after surgery	LEEP
Hildesheim et al., 2016 [23]	Subgroup analysis of an RCT Follow-up 27.3 mo (median)	311 18–25	CIN 1+ CIN 2+	2-valent, 3 doses over 6 months after surgery	LEEP
Ghelardi et al., 2018 [24]	Prospective case-control (SPERANZA project) Follow-up 4 years	344 18–45	CIN 1+ CIN 2+	4-valent at day 30, month 2, and month 6 after surgery	LEEP
Pieralli et al., 2018 [25]	RCT Follow-up 3 years	178 <45	CIN 1+ CIN 2+	4-valent at months 0, 2 and 6 after surgery	Conization (83%), other NA (17%)
Ortega-Quinonero et al., 2019 [24,26]	Retrospective Follow-up 2 years	242 18–65	CIN 2+	2-/4-valent, first dose 0–1 month before or 0–1 month after surgery, other 2 doses over 6 months	LEEP
Sand et al., 2020 [27]	Prospective cohort	17,128 17–51	CIN 2+	2-/4-valent, first dose 0–3 months before or 0–12 months after surgery	Conization
Petrillo et al., 2020 [28]	Retrospective Follow-up 2 years	285 32–47	CIN 1+ CIN 2+	2-/4-valent, first dose 0–1 month after surgery	LEEP
Del Pino et al., 2020 [29]	Prospective Follow up 22.4 mo median	265 26–64	CIN 2+	2-valent at 0, 1 and 6 months after surgery 4-valent at 0, 2 and 6 months after surgery	Conization
Bogani et al., 2020 [30]	Retrospective, multicenter Follow-up 5 years	300 18–89	CIN 2+	2-/4-valent	LEEP
Karimi et al., 2020 [31]	RCT Follow-up 2 years	242 28–36	CIN 1+ CIN 2+	4-valent at months 0, 2 and 6 after conservative treatment	LEEP/Conization
AGWs					
Coskuner et al., 2014 [32]	RCT Follow-up 1 year	171 men 26–42	AGWs	4-valent at months 0, 2 and 6	Electrocautery ± local excision
Joura et al., 2012 [20]	Post-hoc-pooled analysis of 2 RCT (FUTURE I and II) Follow-up 2.5 years (median)	485 15–26	AGWs	4-valent at day 1, month 2, and month 6 after surgery	Surgery
VIN					
Ghelardi et al., 2021 [33]	Prospective case-control Follow-up 2–7 years	118 18–45	VIN	4-valent at day 1, month 2, and month 6 after surgery	Electrosurgical excision and/or laser vaporisation
Joura et al., 2012 [20]	Post-hoc-pooled analysis of 2 RCT (FUTURE I and II) Follow-up 2.5 years (median)	622 15–26	VIN	4-valent at day 1, month 2, and month 6 after surgery	Surgery
AIN					
Swedish et al., 2012 [34]	Retrospective cohort study Follow-up 2 years	202 men who had sex with men 20–72	HGAIN	4-valent at day 1, month 2, and month 6 after surgery	Local excision or ablation
RRP					
Mauz et al., 2018 [35]	Retrospective monocentric study Follow-up 7 years	24 2–48	RRP	4-valent at day 0, week 8, and month 6 after surgery	Microdebridement and intralesional Cidofovir injection

CI, confidence interval; CIN, cervical intraepithelial neoplasia; HGAIN, high-grade anal intraepithelial neoplasia; HPV, human papillomavirus; LEEP, loop electrosurgical excision procedure; NA, not available; RCT, randomized controlled trial; RR, relative risk.

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
