# Peer review of "HPV Vaccination after Primary Treatment of HPV-Related Disease across Different Organ Sites: A Multidisciplinary Comprehensive Review and Meta-Analysis"

_vaccines, 2022, doi:10.3390/vaccines10020239_

Round 1

Reviewer 1 Report

The manuscript by Di Donato et al describes a systematic review of the published data on the effectiveness of using prophylactic HPV vaccines to reduce recurrence in patients treated for HPV-induced disease. This is a subject of considerable debate and the meta analysis supports the therapeutic use of these vaccines, as a safe intervention that results in moderate reduction in recurrence, particularly of CIN. The conclusions also support the need for further study to determine the mechanisms of the protection, which are unlikely to be the same as protection of HPV-naive vaccinees. They further support the need for more analysis of the effectiveness of this intervention in HNC and RP patients.

The data are interesting and support their conclusions, but it would be helpful to the reader if the authors could include a graphic representation of the data, in addition to the detailed, but very dry, tables. 

Reviewer 2 Report

Authors are to be congratulated for an excellent piece of work. The search strategy, inclusion exclusion criteria, PRISMA diagram and risk of bias was addressed. Sensitivity analysis discussed.

One thing that their manuscript does which is not in the DiDonato paper is to address other HPV related diseases. This should be highlighted as value added.

2 Comments

  1. Need to reference DiDonato on line 449
  2. Would having individual data rather than groups of data augment your results? If so this should be mentioned.

Reviewer 3 Report

Violante Di Donato et al. report

a meta-analysis including 21,472 patients from 16 studies, on the efficacy of adjuvant human papillomavirus vaccination on reducing the risk of recurrence of mucosal HPV-associated (pre)malignancies or benign diseases: cervical intraepithelial neoplasia, anal intraepithelial neoplasia, genital warts, and recurrent respiratory papillomatosis... Most of them are related to HPV types covered by 2-/4-valent HPV vaccines.  None of the vaccines has been yet licensed for therapeutic or secondary prevention use.    This meta-analysis confirmed the lower rate of CIN1+, CIN2+, and CIN3+ recurrence in the vaccinated compared to the unvaccinated group, already demonstrated in 2021 by the same team in Vaccines (ref 37). In the present study, the meta-analysis is extended to other HPV-related diseases (genital warts and VIN/VaiN), although there are still few reports (two studies in each disease, respectively) with no significant results.  Nonetheless, the present study is a plea for post-expositional vaccination to reduce the risk of recurrence in older populations. It allows for an informative review of the literature.   

I have minor comments:  

• Regarding the introduction, the authors interestingly consider all the hypotheses and rationale behind the exact protective mechanisms of HPV vaccination in infected individuals. But, one important missing point, that authors could also consider, is the new concept of HPV as a commensal component of the human virome, the presence of high-risk HPV types being proven in samples from the vagina, skin, or gut microbiota of healthy subjects (https://doi.org/10.1128/JVI.00093-14). This observation reinforces a possible mechanism of the secondary prevention of auto-inoculation across new exposed anatomic sites [(b) line 251].  

• In materials and methods, the first sentence (line 263-266) is not clear. “Prophylactic HPV vaccination”, “prophylactic setting” and “as adjuvant” may be better defined here (i.e. primary vs secondary prevention).  

• In materials and methods, section 2.3: Vulvar intraepithelial neoplasia/Vaginal intraepithelial neoplasia "recurrence"? Otherwise, remove “recurrence” in all the lines.  

• Fig.2: I cannot read the end of panel A.   

• Please, replace (35 30.7%) line 422 by (35; 30.7%).  

• The sentence line 433 is not clear: “The RRP occurred in 13 women (54.2%): 2 (15.4%) in the vaccinated and 11 433 (100%) in the unvaccinated cohort”
